# Effects of Starter Cultures and Type of Casings on the Microbial Features and Volatile Profile of Fermented Sausages



Chiara Montanari [1], Federica Barbieri [1,*], Gabriele Gardini [2], Rudy Magnani [2], Davide Gottardi [1,3], Fausto Gardini [1,3] and Giulia Tabanelli [1,3]

[1] Department of Agricultural and Food Sciences, University of Bologna, 40127 Bologna, Italy
[2] C.l.a.i. Soc. Coop., Via Gambellara 62/A, 40026 Imola, Italy
[3] Interdepartmental Centre for Industrial Agri-Food Research, University of Bologna, 47521 Cesena, Italy
[*] Correspondence: federica.barbieri16@unibo.it; Tel.: +39-0547338145

**Abstract:** In the literature, the effect of the type of casing on fermented sausages is quite unexplored, while several studies are focused on the impact of starter cultures. Therefore, this paper studied the effect of three commercial starter cultures and two casings (natural or collagen) on Italian fermented sausages. Physico-chemical parameters ($a_w$, pH, weight loss), microbiota, aroma profile and sensory analysis were evaluated. Results showed that collagen casings promoted a higher reduction of pH and weight loss. Concerning the microbiota, samples with natural casing had higher counts of lactic acid bacteria, while yeast proliferation was promoted in those with collagen. Regardless of the starters and casings applied, levels of enterococci and *Enterobacteriaceae* were low ($\leq 2$ log CFU/g). The aroma profile was significantly affected by casing: despite the starter applied, the presence of collagen casing favoured acid accumulation (mainly acetate and butanoate) and reduction of ketones. Sensory analysis highlighted significant differences only for odour, colour intensity and sourness. The differences observed suggest that collagen casings may provide a greater availability of oxygen. Overall, casings rather than starter cultures impact the microbial and sensorial features of fermented sausages.

**Keywords:** fermented sausages; casings; starter cultures; volatile profile

## 1. Introduction

Fermented meats are among the most important traditional products in Europe. Despite the wide array of products with differences related to geographical and/or cultural issues, some common characteristics are present. Indeed, fermented sausages are generally obtained from minced lean and fat meat (usually pork), supplemented with salt and spices and stuffed into casings obtained from several parts of hog, sheep or beef intestine [1]. Salting has been one of the most important strategies used for food preservation for centuries. The role of salt in sausages is not only to control undesirable microorganisms by lowering the $a_w$, but also to contribute to the textural properties (slice coherency) by impacting the formation of a sol-gel system [2]. Since food industries began aiming to reduce salt for nutritional and health reasons, the role of microorganisms involved in meat fermentations have become more and more important, both for safety issues and technological properties. Lactic fermentation leads to a drop in pH, close to the isoelectric point of meat proteins, with subsequent water loss and ripening phenomena.

The first studies concerning the use of selected starter cultures in sausages were published in the second half of 1950, and only ten years later, the employment of mixed starter cultures containing lactic acid bacteria (LAB) and micrococci was proposed [3]. Initially, their use was mainly aimed to avoid colour defects, control spoilage microorganisms, and shorten the ripening process. In the mid-1990s, the introduction of second-generation starter cultures was implemented in industrial production, mainly focusing on the antagonistic activities against pathogens or toxin-producing bacteria [4,5]. Nowadays, starter

cultures consist of LAB (mainly belonging to the species *Latilactobacillus sakei*, *Lat. curvatus*, *Pediococcus acidilactici*, *P. pentosaceus*) and staphylococci (*Staphylococcus xylosus*, *Staph. carnosus*) isolated from meat [6], together with selected strains of moulds, when required [7]. Other than growth performances, technological properties and competition mechanisms, the selection of starter cultures has been based on the evaluation of their contribution to flavour formation [8], which is an important aspect to guarantee the recognisability of typical products linked to specific geographical areas.

Natural casings are cleaned and left with the submucosa layer by removing mucosa and muscle layers. They are then dry salted, slush salted or cured [9], and therefore must be rehydrated before use. These casings are characterised by tenderness, high water-permeability and elasticity, which allow them to adhere to the meat during ripening. These aspects can be considered an advantage for traditional and high-quality sausages, which are usually hand-stuffed and manufactured. However, they do not guarantee a standardised dimension and form and they are not compatible with industrial automatized stuffing procedures, where standardisation of casings is mandatory [1]. For this reason, several artificial casings, made of cellulose, synthetic polymers, or collagen, are available on the market [10]. Collagen casings contain fibrous and solubilized material obtained from the by-products of the meat industry (hide, bones, connective tissues) [11]. Due to their composition, these products are comparable with the natural ones, being edible and guaranteeing tenderness and cooking characteristics in the final product. In addition, they are uniform in size, standardised and easy to use [1].

While several studies focused on the impact of starter cultures in fermented sausages [12–15], few publications are available regarding the effect of the type of casing on the final product [16–18].

The aim of this paper was to compare the characteristics of Italian fermented sausages obtained using three different commercial starter cultures and two types of sausage casing (natural and collagen). Physico-chemical parameters ($a_w$, pH, weight loss) and microbial populations (LAB, coagulase negative cocci (CNC), yeasts, enterococci, enterobacteria, pseudomonads) were monitored at the end of the fermentation and at the end of ripening. The overall aroma profile of fermented sausages was assessed through SPME-GC-MS (gas chromatography–mass spectrometry coupled with solid phase microextraction) and a sensory analysis of the final products was carried out to highlight the differences related to the starter culture or the type of casing.

## 2. Materials and Methods

### 2.1. Sausage Manufacture

The fermented sausages analysed in this study were Emiliano-type sausages, produced by C.l.a.i. Soc. Coop. (Imola, Italy). For each production, the meat batter (100 kg), consisting of shoulder lean pork meat and minced (3.5 mm) neck pork fat, was mixed with salt (2.5% *w/w*), black pepper (0.16% *w/w*), garlic powder (0.01% *w/w*), dextrose (0.4% *w/w*), ascorbic acid (800 mg/kg), $KNO_3$ (150 mg/kg) and $NaNO_2$ (50 mg/kg). The mixture was divided into three different batches (approx. 30 kg each) into which different commercial starter cultures were inoculated. Each batch was split and stuffed separately into natural (beef middle) or synthetic (collagen) casing with a diameter of 65 mm and an initial weight of approximately 1 kg. The trial was performed in triplicate (three independent sausages for each production). The fermentation and ripening process was carried out at temperatures ranging from 20 to 10 °C, at relative humidity from 65 to 90% for 30 days.

### 2.2. Starter Cultures

During the fermented sausage production process, three different mixtures of commercial starter cultures were added into the meat batter: Bactoferm® SM-181 (*Latilactobacillus sakei* and *Staphylococcus xylosus*; Chr. Hansen, Parma, Italy), Tera Meat® TMX-M (*Lat. sakei*, *Pediococcus acidilactici*, and *Staph. xylosus*; Teracell S.r.l., Cremona, Italy) and Lyocarni SBM-11 (*Lat. sakei*, *Staph. xylosus* and *Staphylococcus carnosus*; Sacco S.r.l., Como, Italy). The initial

concentration of the starter cultures was approx. 7 log CFU/g and 6.10–6.70 log CFU/g for LAB and staphylococci, respectively. A spore suspension of a *Penicillium nalgiovense* strain (Kerry Ingredient, Ireland) was sprayed on casing surfaces.

### 2.3. Physico-Chemical Parameters

The external (approx. 1.5 cm under the casing part of a slice) and internal (central part of a slice) pH were measured using a pH-meter Basic 20 (Crison Instruments, Barcelona, Spain), while $a_w$ was measured with Aqualab 4-TE (Meter Group, Pullman, WA, USA). Each sample was weighed to calculate the mean weight loss (%) with respect to the initial weight. All the analyses were performed in triplicate.

### 2.4. Microbial Counts

After aseptically removing the casing, a slice of approx. 10 g of sample was diluted with 90 mL of 0.9% (*w/v*) NaCl sterile solution in a stomacher bag and homogenised in a Lab Blender Stomacher (Seward Medical, London, UK) for 2 min. Subsequently, decimal dilutions were prepared and plated onto selective media (Oxoid, Basingstoke, UK) for specific microbial enumeration: LAB were counted on de Man-Rogosa-Sharpe Agar and CNC on Mannitol Salt Agar with egg yolk emulsion after 48 h of incubation at 30 °C; yeasts on Sabouraud Dextrose Agar with 200 mg/L of chloramphenicol, after 72 h of incubation at 30 °C; enterococci on Slanetz and Bartley medium incubated for 24 h at 42 °C; *Enterobacteriaceae* on Violet Red Bile Glucose Agar incubated for 24 h at 37 °C; *Pseudomonas* were monitored on Pseudomonas Agar Base with added C-F-C supplement and incubated for 48 h at 30 °C. The analyses were performed in triplicate at initial time (0 days), and after 4 and 30 days.

### 2.5. Aroma Profile Analysis

The aroma profile of samples at the end of ripening (30 days) was analysed through the SPME-GC-MS technique, using Agilent Hewlett-Packard 7890 GC and a 5975C MSD MS detector (Hewlett-Packard, Geneva, Switzerland). Specifically, 3 g of each fermented sausage was placed in 10 mL sterilised vials, added with a known amount of 4-methyl-2-pentanol (Sigma-Aldrich, Steinheim, Germany) as an internal standard, and the identification of the obtained chromatogram was carried out using the Agilent Hewlett–Packard NIST 2011 mass spectral library [19]. Data were expressed as the ratio between the peak area of each compound and the internal standard peak area. The analyses were performed in triplicate.

### 2.6. Sensory Analysis

Sensory analysis was carried out with a trained panel of nine assessors [20]. Thirteen selected descriptors (aroma intensity, bitterness, chewiness, colour intensity and uniformity, elasticity, friability, hardness, juiciness, odour intensity, piquancy, saltiness, sourness) were used to draft the evaluation form, following the order in which they were taken into consideration. Sections were carried out with samples conditioned at 25 °C ($\pm 1$ °C) in a sensory laboratory with individual booths. In each section, the different sausages were evaluated in a randomised order by the panellists.

### 2.7. Statistical Analysis

Data were analysed through a two-way ANOVA model, including a random effect related to the batch. We considered the three different starter cultures together with the use of two different casings as fixed factors. All statistical differences were considered significant at a level of $p \leq 0.05$ using the Bonferroni test. The analyses were performed using the statistical software R [21]. The influence of casing and starter cultures on the final aroma profile was explored by Principal Component Analysis (PCA) using Statistica software (StatSoft Italy srl, Vigonza, Italy).

## 3. Results and Discussion

### 3.1. Physico-Chemical Parameters

The results of pH, $a_w$ and weight loss are reported in Figure 1. With regards to pH, both internal and external values were considered, since in some case the acidification (starting from the inner part of the product due to LAB growth) and the deacidification kinetics (starting from the external part because of oxygen availability and mould activity) can be different, depending on process parameters. In the present study, for each type of casing, no significant differences in relation to the starter culture were observed, while pH was significantly different in relation to the type of casing. The mean pH of the meat mixture used for sausage production was 5.83 ($\pm$0.02). A pH drop of approx. 0.62 and 0.57 units was observed in the internal and external pH of the sausages produced with natural casings after 4 days of fermentation (mean values 5.22 $\pm$ 0.04 and 5.26 $\pm$ 0.05, respectively), without differences related to the starter culture added. At the same time, the pH decrease was higher in samples produced with collagen casing (mean values 5.09 $\pm$ 0.03 and 5.16 $\pm$ 0.03 for internal and external pH, respectively) regardless of the starter culture. At the end of ripening, the overall pH increased but the differences between the two types of sausages were maintained. The internal and external mean pH of sausages with natural casing were 5.45 ($\pm$0.02) and 5.70 ($\pm$0.06), respectively, while those of samples manufactured in collagen casing were 5.22 ($\pm$0.03) and 5.42 ($\pm$0.03). A similar pH behaviour was observed when sausages stuffed into collagen casing were compared with those manufactured in pig casing [18].

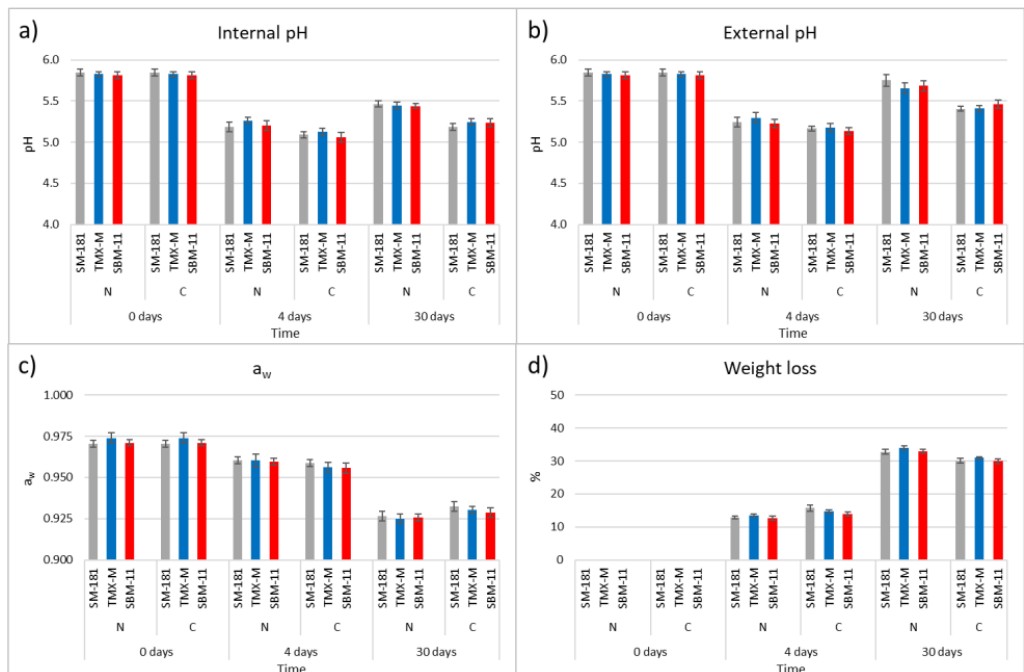

**Figure 1.** Internal pH (**a**), external pH (**b**), $a_w$ (**c**) and weight loss (**d**) monitored on day 0, after 4 days and at the end of ripening (30 days) of fermented sausages obtained using three different starter cultures (SM-181, TMX-M and SBM-11) and stuffed into natural or collagen casings. The data are the mean of three different sausages for each production and standard errors bars are reported.

Concerning $a_w$, the dynamics observed did not show differences in relation to the starter cultures or the type of casing. The initial $a_w$ (0.972 $\pm$ 0.003) was reduced to 0.960 ($\pm$ 0.004) after 4 days, reaching 0.926 ($\pm$0.004) and 0.930 ($\pm$0.003) at the end of ripening for sausages with natural and collagen casings, respectively.

The behaviour of $a_w$ was a consequence of the weight loss, which did not present significant differences (at $p \leq 0.05$) after 4 days (12.97 $\pm$ 0.55% in the sausages with natural casings and 14.77 $\pm$ 0.97% with collagen casings). At the end of ripening, a significantly

higher weight loss characterised the sausages with natural casings ($33.22 \pm 0.78\%$) when compared with those made in collagen ones ($30.33 \pm 0.69\%$). The trends observed for $a_w$ and weight loss were comparable with those reported in literature for similar fermented sausages [22,23].

### 3.2. Microbial Counts

Microbial counts are reported in Table 1, together with the results of a two-way ANOVA to better highlight the effects of casings, starter cultures, and their interactions on the microbiota. All the models showed a negligible variance component related to the production. Pseudomonads are not reported because their concentration was always below the detection limit (<1 log CFU/g).

At 0 days, the counts of autochthonous microorganisms (yeasts, enterococci, *Enterobacteriaceae*) did not show significant differences in relation to starter cultures and type of casing. Starter cultures were added at approx. 7 log CFU/g in the meat batter. Regardless of the type of casing, sausages with TMX-M were characterised by a significant lower concentration of CNC (approx. 0.5 log units).

After 4 days, when the first fermentation step was completed, LAB counts were always higher than 8 log CFU/g and the counts were significantly influenced both by starter cultures and casings. The sausages inoculated with TMX-M reached significantly higher LAB counts, compared to SM-181. In addition, LAB counts were always significantly higher in samples produced in natural casing. CNC presented different concentrations and were influenced by the two factors and their interaction. Their counts were generally significantly higher in samples with collagen casings. Sausages obtained with TMX-M and natural casing presented the lowest CNC counts. Mean yeast counts were significantly higher in collagen-casing sausages even though they did not exceed 3 log CFU/g. Similar considerations can be extended to *Enterobacteriaceae*, whose concentration was always lower than 2 log CFU/g. The samples produced with collagen casings were characterised by slightly, but significantly, higher counts of enterococci, with the exception of sausages inoculated with SBM-11.

Higher LAB counts found in sausages produced in natural casings did induce a higher pH reduction. The pH of samples manufactured in collagen casing was lower at 4 days. This could be explained by a different equilibrium in the fate of pyruvate when fermentable sugars are limited. Pyruvate may derive from glycolysis, but it can also be produced by LAB from the metabolism of some amino acids and used as an energy source [24].

At the end of ripening, LAB counts were affected by the type of casings and the interaction between casing and starter culture, but not by starter culture alone. In samples with natural casing, LAB counts ranged between 8.44 and 8.71 log CFU/g, while the counts were always lower than 8 log CFU/g (from 7.51 to 7.64 log CFU/g) in sausages with collagen casings. Another study [21] did not show significant differences between LAB at the end of ripening in relation to the type of casing; however in that study, the sausages reached an $a_w$ value of 0.8 in few days, which limits the metabolic activities of these bacteria. The arginine deiminase (ADI) pathway is essential to explain the metabolic activity and survival of *Lat. sakei* in meat environments when fermentable sugars are depleted [25]. Its activation is favoured by low oxygen and glucose concentration and presence of ribose [26].

At the end of ripening, CNC counts ranged from 7.24 to 7.81 log CFU/g and their concentration was affected by the starter cultures and their interaction with the casing. The highest CNC count was observed in sausages with SM-181 and where collagen casings were used, indicating a greater O2 that favours CNC growth and inhibits ADI activity in LAB.

**Table 1.** Microbial counts (expressed as log CFU/g) on day 0, after 4 days and at the end of ripening (30 days) of fermented sausages obtained using three different starter cultures (SM-181, TMX-M and SBM-11) and stuffed into two different casings (natural and collagen). The data are the mean of three determinations in three independent samples (n = 9) and standard errors are reported. The results of a two-way ANOVA are also reported to evaluate the effects of the different experimental conditions and their interactions on counts. Differences were considered significant at $p \leq 0.05$.

| Casing | Microbial Group | 0 Days Starter SM-181 | 0 Days Starter TMX-M | 0 Days Starter SBM-11 | 0 Days p-Value C | 0 Days p-Value S | 0 Days p-Value C × S | 4 Days Starter SM-181 | 4 Days Starter TMX-M | 4 Days Starter SBM-11 | 4 Days p-Value C | 4 Days p-Value S | 4 Days p-Value C × S | 30 Days Starter SM-181 | 30 Days Starter TMX-M | 30 Days Starter SBM-11 | 30 Days p-Value C | 30 Days p-Value S | 30 Days p-Value C × S |
|---|---|---|---|---|---|---|---|---|---|---|---|---|---|---|---|---|---|---|---|
| Natural | Lactic acid bacteria | 6.88 ± 0.03 | 7.01 ± 0.03 | 7.00 ± 0.03 | NS † | NS | NS | 8.21 [a] ± 0.02 | 8.84 [b] ± 0.02 | 8.62 [c] ± 0.02 | ** | ** | NS | 8.44 [a] ± 0.02 | 8.43 [a] ± 0.03 | 8.71 [b] ± 0.02 | ** | NS | * |
| Collagen | | 6.97 ± 0.08 | 7.08 ± 0.05 | 6.94 ± 0.06 | | | | 8.02 [d] ± 0.03 | 8.45 [e] ± 0.02 | 8.28 [a] ± 0.03 | | | | 7.52 [c] ± 0.02 | 7.64 [c] ± 0.02 | 7.50 [c] ± 0.03 | | | |
| Natural | Coagulase negative cocci | 6.63 [a] ± 0.03 | 6.08 [b] ± 0.02 | 6.77 [a] ± 0.02 | NS | ** | NS | 6.30 [a] ± 0.02 | 5.87 [b] ± 0.03 | 6.32 [ac] ± 0.02 | ** | ** | ** | 7.40 [a] ± 0.01 | 7.39 [a] ± 0.02 | 7.51 [a] ± 0.02 | NS | ** | ** |
| Collagen | | 6.68 [a] ± 0.03 | 6.10 [b] ± 0.05 | 6.70 [a] ± 0.02 | | | | 6.95 [d] ± 0.01 | 7.48 [e] ± 0.01 | 6.48 [c] ± 0.02 | | | | 7.81 [b] ± 0.01 | 7.42 [a] ± 0.03 | 7.24 [c] ± 0.03 | | | |
| Natural | Yeasts | 1.54 ± 0.04 | 1.51 ± 0.01 | 1.68 ± 0.05 | NS | NS | NS | 2.70 [a] ± 0.02 | 2.00 [b] ± 0.01 | 1.23 [c] ± 0.03 | ** | * | * | 1.96 [a] ± 0.04 | 2.00 [a] ± 0.01 | 1.35 [b] ± 0.04 | ** | ** | ** |
| Collagen | | 1.52 ± 0.03 | 1.51 ± 0.04 | 1.67 ± 0.04 | | | | 2.78 [ad] ± 0.03 | 2.96 [d] ± 0.04 | 2.00 [b] ± 0.01 | | | | 6.15 [c] ± 0.04 | 6.79 [d] ± 0.01 | 6.50 [e] ± 0.01 | | | |
| Natural | Enterococci | 2.32 ± 0.03 | 2.20 ± 0.02 | 2.49 ± 0.03 | NS | NS | NS | 2.05 [a] ± 0.05 | 2.05 [a] ± 0.06 | 2.01 [a] ± 0.09 | ** | NS | NS | 2.07 [ad] ± 0.03 | 2.09 [ad] ± 0.03 | 2.38 [b] ± 0.05 | ** | ** | NS |
| Collagen | | 2.27 ± 0.08 | 2.26 ± 0.03 | 2.45 ± 0.05 | | | | 2.37 [b] ± 0.05 | 2.65 [b] ± 0.06 | 1.49 [c] ± 0.05 | | | | 1.89 [ac] ± 0.04 | 1.81 [c] ± 0.04 | 2.26 [bd] ± 0.05 | | | |
| Natural | Enterobacteria | 1.95 ± 0.06 | 2.00 ± 0.04 | 1.81 ± 0.02 | NS | NS | NS | 1.00 [a] ± 0.07 | 1.29 [ab] ± 0.04 | 1.02 [a] ± 0.05 | ** | ** | * | 2.16 [a] ± 0.05 | 2.15 [a] ± 0.03 | 1.79 [b] ± 0.07 | ** | * | ** |
| Collagen | | 1.95 ± 0.06 | 2.00 ± 0.03 | 1.81 ± 0.03 | | | | 1.60 [ab] ± 0.06 | 1.77 [b] ± 0.06 | 1.00 [a] ± 0.05 | | | | 1.30 [c] ± 0.07 | 1.86 [b] ± 0.09 | 1.48 [c] ± 0.05 | | | |

NS †: not significant; *: *p*-Value between 0.05 and 0.01; **: *p*-Value < 0.01. Different letters within the rows indicate differences ($p \leq 0.05$).

Yeasts showed significant differences in relation to the casing applied. In fact, while their number did not increase in sausages with natural casing, concentrations higher than 6 log CFU/g were reached in all the samples with collagen casing. The presence of yeast in sausages is well-documented and their cell load in ripened products can be extremely variable. Their role concerns oxygen consumption, production of lipolytic and proteolytic enzymes and, in turn, flavour fingerprinting [27]. The most important species (*Debaryomyces hansenii*, *Yarrowia lipolytica*) are aerobic or weakly fermentative [28–30], suggesting a major $O_2$ permeability of collagen casings.

Enterobacteria and enterococci did not exhibit any notable growth at any time, regardless of the starter cultures and the type of casing. In addition, the final concentrations of enterococci and *Enterobacteriaceae* were low, and depended on the quality of raw meat and the rapid and effective colonisation of the starter cultures [4,31].

### 3.3. Flavour Formation

The volatile molecule profile of sausages at the end of ripening was analysed through a SPME-GC-MS protocol. A total of 41 volatile molecules were identified and the data are reported in Table 2. The molecules were grouped into aldehydes, ketones, alcohols and acids according to their chemical structure.

Figure 2 represents the percentage of these chemical groups with respect to the starter cultures and the casings adopted. The presence of TMX-M was responsible for higher acid accumulation, while SM-181 produced higher amounts of ketones. The percentage of alcohols and aldehydes was fairly constant. In addition, these results clearly showed the effect of casing on volatile profile: in fact, the collagen casing favoured the production of acids and decreased the percentage of ketones, regardless of the starter cultures adopted.

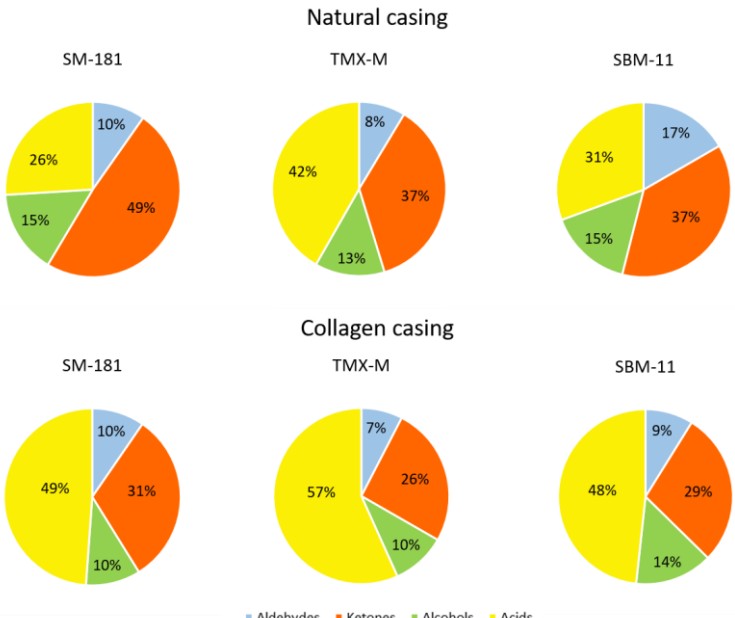

**Figure 2.** Percentage of volatile compounds grouped based on their chemical structure (aldehydes, ketones, alcohols and acids) detected in the different samples at the end of ripening.

Aldehydes were mainly represented by nonanal and decanal, which are aliphatic aldehydes deriving from autoxidation of unsaturated fatty acids. They are responsible for green notes, although their excessive accumulation can result in a rancid flavour [32]. Benzaldehyde and benzenacetaldehyde derive from the metabolism of aromatic amino acids and they confer almond and fruity notes [27]. However, there were no marked differences concerning aldehydes in relation to starter cultures or casings, apart from a higher content of nonanal in sausages inoculated with SBM-11 and manufactured in natural casings.

**Table 2.** Volatile organic compounds (VOCs) of ripened fermented sausages detected by SPME-GC-MS, expressed as a ratio between the peak area of each molecule and the peak area of the internal standard. The results of a two-way ANOVA are also reported to evaluate the effects of the different experimental conditions and their interactions on VOCs accumulation. Differences were considered significant at $p \leq 0.05$.

| | Natural Casing | | | Collagen Casing | | |
|---|---|---|---|---|---|---|
| Compounds | SM-181 | TMX-M | SBM-11 | SM-181 | TMX-M | SBM-11 |
| Butanal | 0.01 ± 0.00 | 0.03 ± 0.01 | 0.03 ± 0.00 | 0.05 ± 0.00 | 0.04 ± 0.00 | 0.03 ± 0.00 |
| Hexanal | 0.25 ± 0.09 | 0.41 ± 0.06 | 0.50 ± 0.02 | 0.76 ± 0.05 | 0.49 ± 0.14 | 0.80 ± 0.07 |
| Nonanal | 2.09 [a] ± 0.18 | 2.49 [a] ± 0.05 | 6.16 [b] ± 0.76 | 2.49 [a] ± 0.23 | 1.79 [a] ± 0.17 | 1.76 [a] ± 0.02 |
| Decanal | 0.61 [a] ± 0.26 | 0.78 [a] ± 0.16 | 0.84 [a] ± 0.10 | 1.19 [b] ± 0.21 | 1.30 [ab] ± 0.05 | 1.26 [ab] ± 0.05 |
| Benzaldehyde | 0.26 ± 0.10 | 0.31 ± 0.02 | 0.39 ± 0.04 | 0.45 ± 0.08 | 0.48 ± 0.02 | 0.36 ± 0.02 |
| Benzeneacetaldehyde | 0.60 ± 0.14 | 0.43 ± 0.01 | 0.58 ± 0.07 | 0.44 ± 0.06 | 0.37 ± 0.03 | 0.44 ± 0.01 |
| **ALDEHYDES** | **3.82 [a] ± 0.10** | **4.45 [b] ± 0.16** | **8.50 [b] ± 0.93** | **5.38 [ac] ± 0.37** | **4.48 [a] ± 0.38** | **4.65 [a] ± 0.04** |
| Acetone | 4.04 [a] ± 0.11 | 3.39 [b] ± 0.12 | 3.29 [b] ± 0.17 | 6.21 [c] ± 0.43 | 4.02 [a] ± 0.06 | 4.70 [d] ± 0.31 |
| 2-butanone | 2.64 [a] ± 0.16 | 1.41 [b] ± 0.07 | 1.73 [b] ± 0.07 | 0.68 [c] ± 0.04 | 0.46 [c] ± 0.01 | 0.78 [c] ± 0.12 |
| 2-pentanone | 0.37 [a] ± 0.02 | 0.27 [a] ± 0.01 | 0.32 [a] ± 0.04 | 0.53 [ab] ± 0.05 | 0.45 [a] ± 0.03 | 0.60 [c] ± 0.06 |
| 2,3-butanedione | 0.89 [a] ± 0.13 | 1.08 [a] ± 0.17 | 0.87 [a] ± 0.16 | 0.47 [b] ± 0.03 | 0.29 [b] ± 0.08 | 0.15 [b] ± 0.02 |
| Methyl isobutil ketone | 1.30 [a] ± 0.02 | 1.08 [b] ± 0.03 | 1.14 [b] ± 0.00 | 1.31 [a] ± 0.06 | 1.42 [a] ± 0.11 | 1.37 [a] ± 0.03 |
| 5-methyl-3-hexanone | 0.30 ± 0.02 | 0.20 ± 0.03 | 0.11 ± 0.02 | 0.38 ± 0.01 | 0.12 ± 0.04 | 0.25 ± 0.03 |
| 4-methyl-3-penten-2-one | 4.07 [a] ± 0.98 | 5.43 [b] ± 0.11 | 5.68 [b] ± 0.11 | 5.53 [b] ± 0.12 | 6.07 [bc] ± 0.05 | 5.70 [c] ± 0.11 |
| 2-heptanone | 1.30 [a] ± 0.13 | 1.27 [a] ± 0.05 | 1.32 [a] ± 0.08 | 0 † [bc] | 0.15 [c] ± 0.09 | 0.44 [c] ± 0.03 |
| 2-octanone | 0.75 [a] ± 0.11 | 0.90 [a] ± 0.21 | 0.87 [a] ± 0.03 | 0.59 [ab] ± 0.02 | 0.34 [b] ± 0.10 | 0.18 [b] ± 0.10 |
| 3-hydroxy-2-butanone | 2.84 [a] ± 0.36 | 3.40 [a] ± 0.59 | 2.86 [a] ± 0.12 | 1.55 [b] ± 0.03 | 1.54 [b] ± 0.50 | 0.47 [b] ± 0.03 |
| 2-nonanone | 0.43 [a] ± 0.07 | 0.33 [a] ± 0.01 | 0.67 [a] ± 0.07 | 0.17 [b] ± 0.02 | 0.14 [b] ± 0.01 | 0.14 [b] ± 0.01 |
| Acetophenone | 0.17 ± 0.03 | 0.13 ± 0.01 | 0.16 ± 0.01 | 0.25 ± 0.02 | 0.14 ± 0.02 | 0.14 ± 0.01 |
| **KETONES** | **19.10 [a] ± 1.13** | **18.88 [a] ± 1.05** | **19.02 [a] ± 0.33** | **17.66 [ab] ± 0.63** | **15.14 [b] ± 0.37** | **14.92 [b] ± 0.38** |
| Isopropyl alcohol | 0.45 [a] ± 0.02 | 0.41 [a] ± 0.02 | 0.39 [a] ± 0.02 | 0.31 [b] ± 0.03 | 0.29 [b] ± 0.00 | 0.30 [b] ± 0.02 |
| Ethyl alcohol | 2.50 [a] ± 0.15 | 3.02 [a] ± 0.12 | 4.08 [b] ± 0.20 | 2.47 [a] ± 0.25 | 3.14 [c] ± 0.08 | 4.63 [b] ± 0.62 |
| 2-butanol | 0.41 [a] ± 0.02 | 0.47 [a] ± 0.01 | 0.36 [a] ± 0.03 | 0 [b] | 0 [b] | 0 [b] |
| 1-pentanol | 0.16 ± 0.01 | 0.22 ± 0.02 | 0.25 ± 0.03 | 0.20 ± 0.02 | 0.18 ± 0.01 | 0.21 ± 0.01 |
| 3-methyl-3-buten-1-ol | 0.17 ± 0.01 | 0.12 ± 0.01 | 0.08 ± 0.01 | 0.35 ± 0.02 | 0.09 ± 0.03 | 0.16 ± 0.01 |
| 3-methyl-2-buten-1-ol | 0.16 [a] ± 0.02 | 0.35 [b] ± 0.02 | 0.30 [b] ± 0.06 | 0.35 [b] ± 0.02 | 0.52 [bc] ± 0.03 | 0.57 [c] ± 0.05 |
| 1-hexanol | 0 [a] | 0.50 [b] ± 0.03 | 0.52 [b] ± 0.04 | 0.48 [b] ± 0.07 | 0.27 [b] ± 0.08 | 0.37 [b] ± 0.04 |
| 1-octen-3-ol | 1.54 [a] ± 0.20 | 0.88 [b] ± 0.05 | 0.99 [ab] ± 0.03 | 0.60 [b] ± 0.04 | 0.41 [c] ± 0.01 | 0.42 [c] ± 0.02 |
| 1-octanol | 0.28 [a] ± 0.01 | 0.36 [a] ± 0.03 | 0.71 [a] ± 0.07 | 0.26 [b] ± 0.01 | 0.24 [a] ± 0.02 | 0.20 [a] ± 0.01 |
| Benzyl alcohol | 0.14 [a] ± 0.00 | 0.27 [b] ± 0.02 | 0.20 [ab] ± 0.03 | 0.33 [b] ± 0.03 | 0.42 [c] ± 0.02 | 0.36 [b] ± 0.00 |
| Phenylethyl alcohol | 0.24 [a] ± 0.02 | 0.10 [a] ± 0.05 | 0 [b] | 0.25 [a] ± 0.02 | 0.34 [a] ± 0.03 | 0.33 [a] ± 0.04 |
| **ALCOHOLS** | **6.05 ± 0.09** | **6.68 ± 0.23** | **7.88 ± 0.46** | **5.59 ± 0.48** | **5.89 ± 0.21** | **7.54 ± 0.78** |
| Acetic acid | 6.43 [a] ± 0.20 | 14.31 [b] ± 1.23 | 8.14 [a] ± 1.05 | 15.37 [a] ± 1.57 | 19.42 [bc] ± 2.25 | 12.61 [b] ± 0.65 |
| Butanoic acid | 1.07 [a] ± 0.02 | 2.29 [b] ± 0.31 | 1.13 [a] ± 0.16 | 3.88 [c] ± 0.35 | 4.54 [c] ± 0.32 | 3.93 [c] ± 0.28 |
| 3-methyl-Butanoic acid | 0 [a] | 0 [a] | 0 [a] | 0.33 [b] ± 0.03 | 1.13 [c] ± 0.07 | 1.35 [c] ± 0.07 |
| Pentanoic acid | 0 [a] | 0 [a] | 0 [a] | 0 [a] | 0.20 [b] ± 0.06 | 0.26 [b] ± 0.01 |
| Hexanoic acid | 0.69 [a] ± 0.05 | 0.91 [a] ± 0.05 | 0.76 [a] ± 0.08 | 1.30 [b] ± 0.14 | 1.32 [b] ± 0.11 | 1.13 [ab] ± 0.09 |
| Heptanoic acid | 0.25 ± 0.01 | 0.26 ± 0.02 | 0.23 ± 0.02 | 0.34 ± 0.05 | 0.31 ± 0.02 | 0.30 ± 0.03 |
| Octanoic acid | 0.68 [a] ± 0.02 | 0.91 [a] ± 0.06 | 0.74 [a] ± 0.08 | 1.58 [b] ± 0.18 | 1.62 [b] ± 0.12 | 1.34 [b] ± 0.02 |
| Nonanoic acid | 0.57 [a] ± 0.02 | 0.85 [a] ± 0.05 | 0.82 [a] ± 0.10 | 1.12 [b] ± 0.12 | 0.88 [a] ± 0.05 | 1.21 [b] ± 0.12 |
| n-decanoic acid | 0.48 [a] ± 0.02 | 0.76 [a] ± 0.11 | 0.77 [a] ± 0.09 | 0.97 [ab] ± 0.11 | 1.27 [b] ± 0.06 | 1.22 [b] ± 0.08 |
| n-hexadecanoic acid | 0 [a] | 0 [a] | 0.90 [b] ± 0.52 | 0 [a] | 1.03 [b] ± 0.30 | 0.61 [b] ± 0.35 |
| Dodecanoic acid | 0 [a] | 0.62 [b] ± 0.18 | 1.00 [b] ± 0.41 | 1.36 [bc] ± 0.24 | 0.49 [b] ± 0.29 | 0.41 [b] ± 0.12 |
| Tetradecanoic acid | 0 [a] | 0.59 [b] ± 0.21 | 1.11 [c] ± 0.22 | 1.11 [c] ± 0.02 | 1.22 [c] ± 0.10 | 0.93 [ac] ± 0.11 |
| **ACIDS** | **10.16 [a] ± 0.21** | **21.50 [b] ± 2.01** | **15.61 [ab] ± 1.90** | **27.35 [b] ± 2.62** | **33.43 [bc] ± 2.85** | **25.31 [b] ± 1.32** |

† below the detection limit (0.1). Different letters within the rows indicate differences ($p \leq 0.05$).

Among ketones, methyl ketones (such as 2-heptanone, 2-octanone and 2-nonanone) have considerable impact on the sausage's aroma profile. They can be formed through the β-oxidation of fatty acids by microorganisms, particularly moulds and staphylococci [33]. Other important ketones, such as 2,3-butandione (diacetyl) and 3-hydroxy-2-butanone (acetoin), can derive mainly from LAB pyruvate metabolism [33,34]. Their concentration was higher in sausages with natural casings, where LAB showed the best survival, confirming the results of Yan et al. [18]. Pyruvate can also be the precursor of 2-butanone, 2-pentanone, acetone and butanoic acid [8,35].

The presence of alcohols was low in all the samples. The major constituent of this group was ethanol, which can be the result of several LAB and staphylococci pathways, such as pyruvate or amino acid metabolism [27,36,37].

As already observed, acid concentration was higher in the samples with collagen casing. Similar behaviour with respect to casings was observed by Yan et al. [18]. The starter culture TMX-M was responsible for the highest acid accumulation, regardless of casings applied. As expected, acetate was the most abundant acid. Acetate is the end-product of several pathways. Particularly, it can be produced by LAB as a product of mixed-acid fermentation, activated when fermentable sugars are limited or absent. The accumulation of acetate can follow two routes: the pyruvate formate lyase (favoured by the absence of $O_2$) and the pyruvate oxidase (activated by $O_2$ availability) [38]. In the first route, there is an equimolar production of both ethanol and acetate. However, no ethanol increase was observed in those sausages where acetate was higher. In the second route, acetate production occurs with $H_2O_2$ release, that can then induce possible discolorations and oxidation. However, the presence of high concentrations of microorganisms possessing catalase activity (in particular staphylococci and yeasts) could avoid these processes. The sausages produced in collagen casing presented the highest concentration of other acids, such as butanoic (butyric), hexanoic, octanoic, and decanoic acid. These acids have been found in sausages and were described for their contribution to the final aroma [33,39]. Among these acids, butanoic acid was the most abundant. Its presence in pork fat has not been reported. However, similar to other compounds such as diacetyl, acetoin, 2-butanone, 2-butanol, it can be produced via pyruvate metabolism [8,40]. In addition, 3-methyl-butanoic (isovaleric) acid was detected only in samples with collagen casings. It derives from the oxidative decarboxylation of leucine and its amount can negatively or positively impact the final sensorial properties of sausages [41].

The influence of casing and starter cultures on the final aroma profile of fermented sausages was explored by principal component analysis (PCA), the results of which are represented in Figure 3. Factor 1 (PC1) explained 73.34% of the variability, while Factor 2 (PC2) explained 11.18%.

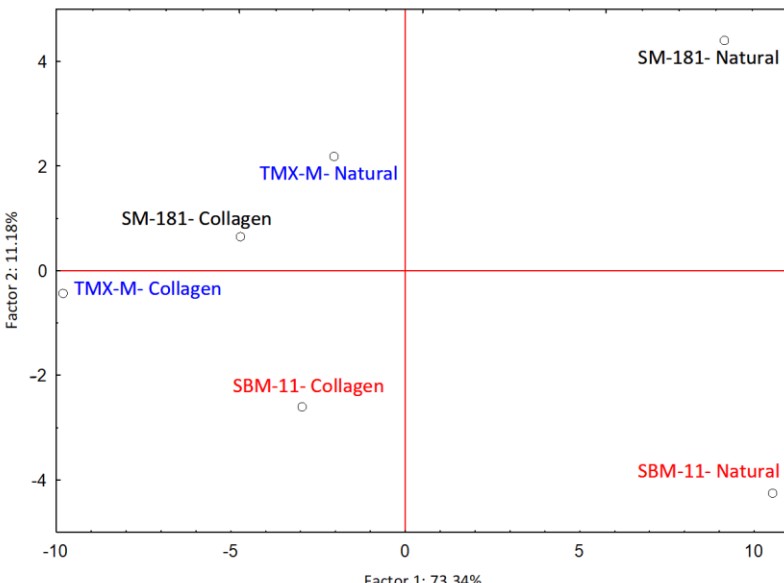

**Figure 3.** PCA case coordinates for the first two factors explaining the influence of adopted casing and starter cultures on the final aroma profile of fermented sausages.

Table 3 reports the 41 molecules whose variable contribution to the covariates was higher than 0.02 in at least one of the first two factors of the PCA, as well as the factor coordinates of the variables. The main contributor to Factor 1 is acetic acid (0.637) with a factor coordinate negative (-6.43), followed by nonanal (0.099) with a positive factor

coordinate (2.54). Butanoic acid (the third in order of contribution, 0.079) was characterised, similarly to acetic acid, by a negative factor coordinate (−2.26). The sausages produced with collagen casing were grouped on the left part of the graph due to the high acid content. Moreover, sausages produced with the starter culture TMX-M, which was responsible for the highest acetate accumulation, were also located on the left side of the graph, even if well-separated according to the type of casing adopted (Figure 3). The most important contributor to Factor 2 was nonanal (0.214 with a negative factor coordinate −1.46), which contributed to the separation of the sausages fermented with SBM-11 in natural casings from the others. Acetoin (0.157), 2-butanone (0.138) and acetone (0.081), all deriving mainly from LAB pyruvate metabolism, were among the most important contributors to Factor 2, both with positive factor coordinates (Table 3).

**Table 3.** Variable contribution and relative factor coordinates of the volatile compounds for PC1 and PC2. Compounds with higher contribution (>0.05) to at least one of the first two Principal Components are reported in bold.

| Volatile Compound | Variable Contribution, Based on Covariances, to PC1 and PC2 | | Factor Coordinates of the Variables PC1 and PC2, Based on Correlations | |
|---|---|---|---|---|
| | PC1 | PC2 | PC1 | PC2 |
| Butanal | 0.0000 | 0.0000 | −0.0133 | −0.0041 |
| Hexanal | 0.0002 | 0.0035 | −0.1384 | −0.1888 |
| **Nonanal** | **0.0994** | **0.2143** | **2.5428** | **−1.4578** |
| Decanal | 0.0011 | 0.0032 | −0.2767 | −0.1782 |
| Benzaldehyde | 0.0000 | 0.0001 | −0.0320 | −0.0362 |
| Benzeneacetaldehyde | 0.0011 | 0.0014 | 0.2727 | 0.1212 |
| **Acetone** | **0.0000** | **0.0812** | **0.0271** | **0.8975** |
| **2-butanone** | **0.0524** | **0.1380** | **1.8463** | **1.1698** |
| 2-pentanone | 0.0000 | 0.0000 | −0.0374 | 0.0023 |
| **2,3-butanedione** | **0.0062** | **0.0161** | **0.6368** | **0.4006** |
| Methyl isobutyl ketone | 0.0004 | 0.0044 | 0.1651 | 0.2108 |
| 5-methyl-3-hexanone | 0.0000 | 0.0022 | 0.0469 | 0.1503 |
| 4-methyl-3-penten-2-one | 0.0002 | 0.0159 | 0.1339 | −0.3979 |
| 2-heptanone | 0.0214 | 0.0182 | 1.1797 | 0.4253 |
| 2-octanone | 0.0037 | 0.0082 | 0.4960 | 0.2864 |
| **3-hydroxy-2-butanone** | **0.0492** | **0.1567** | **1.7891** | **1.2468** |
| 2-nonanone | 0.0030 | 0.0000 | 0.4482 | −0.0144 |
| Acetophenone | 0.0000 | 0.0002 | 0.0328 | 0.0458 |
| Isopropyl alcohol | 0.0005 | 0.0020 | 0.1894 | 0.1419 |
| **Ethyl alcohol** | **0.0094** | **0.0859** | **0.7830** | **−0.9231** |
| 2-butanol | 0.0021 | 0.0056 | 0.3741 | 0.2356 |
| 1-pentanol | 0.0000 | 0.0000 | 0.0527 | −0.0162 |
| 3-methyl-3-buten-1-ol | 0.0000 | 0.0006 | −0.0152 | 0.0821 |
| 3-methyl-2-buten-1-ol | 0.0004 | 0.0012 | −0.1621 | −0.1109 |
| 1-hexanol | 0.0000 | 0.0053 | −0.0533 | −0.2297 |
| 1-octen-3-ol | 0.0137 | 0.0486 | 0.9453 | 0.6946 |
| 1-octanol | 0.0014 | 0.0016 | 0.3028 | −0.1268 |
| Benzyl alcohol | 0.0002 | 0.0001 | −0.1349 | −0.0370 |
| Phenylethyl alcohol | 0.0001 | 0.0008 | −0.1066 | 0.0928 |
| **Acetic acid** | **0.6365** | **0.0274** | **−6.4331** | **0.5217** |
| **Butanoic acid** | **0.0786** | **0.0136** | **−2.2613** | **−0.3680** |
| 3-methyl-Butanoic acid | 0.0081 | 0.0206 | −0.7272 | −0.4525 |
| Pentanoic acid | 0.0002 | 0.0008 | −0.1270 | −0.0939 |
| Hexanoic acid | 0.0010 | 0.0000 | −0.2654 | 0.0298 |
| Heptanoic acid | 0.0000 | 0.0001 | −0.0009 | 0.0403 |
| Octanoic acid | 0.0039 | 0.0001 | −0.5039 | −0.0380 |
| Nonanoic acid | 0.0001 | 0.0012 | −0.1102 | −0.1134 |
| n-decanoic acid | 0.0014 | 0.0050 | −0.3052 | −0.2226 |
| **n-hexadecanoic acid** | **0.0000** | **0.0501** | **−0.0703** | **−0.7055** |
| Dodecanoic acid | 0.0001 | 0.0148 | −0.1090 | −0.3840 |
| Tetradecanoic acid | 0.0022 | 0.0489 | −0.3815 | −0.6966 |

### 3.4. Sensory Analysis

Sensory analysis was carried out by a trained panel (9 participants) considering 13 descriptors [23]. The mean score values for each attribute are reported in Figure 4.

Some of the attributes (bitterness, piquancy, hardness, chewiness) showed small variations among the samples, while others were characterised by marked differences.

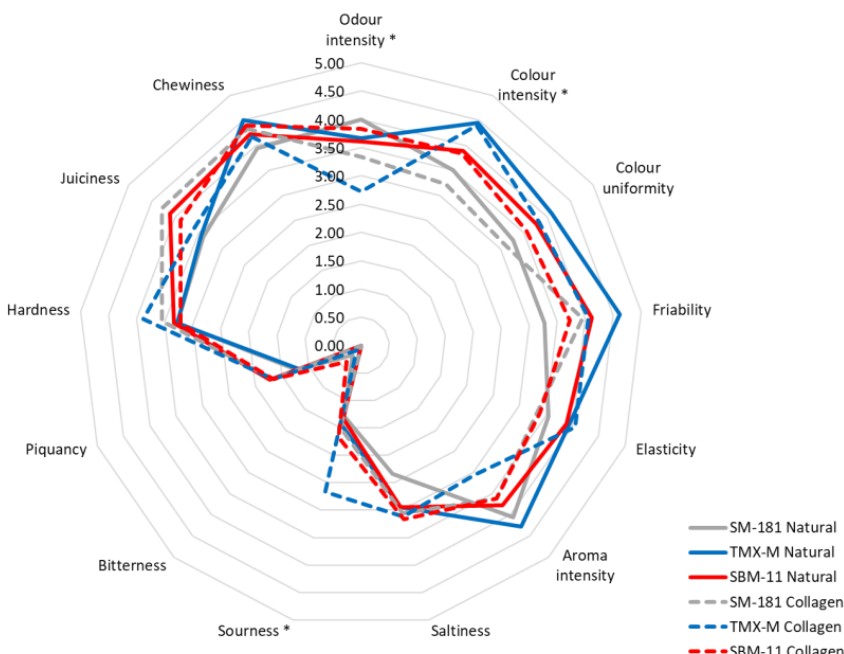

**Figure 4.** Sensory data attributed, at the end of ripening, to fermented sausages produced using different starter cultures and stuffed into different casings. For each attribute, the presence of an asterisk indicates significant differences between the samples.

However, a two-way ANOVA applied to all the attributes indicated that only odour intensity, colour intensity and sourness were significant ($p \leq 0.05$) (Table 4). With regards to odour intensity, only casing presented $p < 0.05$, since the scores for this attribute were lower in the samples with collagen casing and inoculated with SM-181 and TMX-M. On the other hand, the starter culture significantly influenced colour intensity ($p < 0.01$). TMX-M presented the highest score for this attribute, regardless of the casing applied, while SM-181 received the lowest one. The sourness attribute was significantly influenced by casing: all the sausages with collagen casings received higher scores, with the highest one observed in samples containing TMX-M. These values confirmed the instrumental data, which revealed a higher acid accumulation (and a lower pH) in the presence of collagen casings.

**Table 4.** Sensory analysis attributes characterised by significant variations in relation to the type of samples (odour intensity, colour intensity and sourness). The mean values of the scores for each attribute are reported. Also, standard errors are reported. The results of a two-way ANOVA are also reported to evaluate the effects of the different experimental conditions and their interactions on sensory evaluation. Differences were considered significant at $p \leq 0.05$.

| Casing | Attribute | Starter SM-181 | Starter TMX-M | Starter SBM-11 | p-Value C | p-Value S | p-Value C × S |
|---|---|---|---|---|---|---|---|
| Natural | Odour intensity | 4.00 [a] ± 0.08 | 3.67 [ab] ± 0.10 | 3.61 [ab] ± 0.11 | * | NS † | NS |
| Collagen | | 3.22 [bc] ± 0.10 | 2.72 [c] ± 0.07 | 3.67 [ab] ± 0.08 | | | |
| Natural | Colour intensity | 3.50 [a] ± 0.10 | 4.44 [b] ± 0.10 | 3.89 [ab] ± 0.10 | NS | ** | NS |
| Collagen | | 3.22 [a] ± 0.10 | 4.39 [b] ± 0.11 | 3.83 [ab] ± 0.08 | | | |
| Natural | Sourness | 1.28 [a] ± 0.13 | 1.44 [a] ± 0.06 | 1.33 [a] ± 0.11 | * | NS | NS |
| Collagen | | 1.50 [a] ± 0.11 | 2.67 [b] ± 0.11 | 1.67 [a] ± 0.08 | | | |

NS †: not significant; *: *p*-value between 0.05 and 0.01; **: *p*-value < 0.01. Different letters within the rows indicate differences ($p \leq 0.05$).

## 4. Conclusions

Despite the widespread use of collagen casings in the cured meat industry [9], little information is available on the impact that they have on fermented sausages. Most of these studies are focused on their mechanical properties rather than their permeability. Conte et al. [16] described better textural properties of fermented sausages from bovine meat prepared with natural casings compared to those with collagen casings. The differences observed in our work seem to indicate a greater availability of oxygen in the sausages stuffed into collagen casings. However, to the best of our knowledge, there are no studies describing the different permeability to oxygen among sausage casings. Therefore, the latter consideration remains a hypothesis indirectly supported by the microbial population dynamics and the composition of the aroma profile. The growth of yeasts observed in collagen casings seems to suggest that this environmental niche was colonized by these fungi instead of moulds, changing some relevant characteristics of the final products. Overall, casings rather than starter cultures affected the microbial and sensorial features of fermented sausages. Based on these preliminary indications, future research activities will be aimed to also better elucidate the effects of collagen casing on the chemical features of fermented sausages, such as the modification of lipid fractions and proteolysis that can play a role in the formation of the final product aroma profile.

**Author Contributions:** Conceptualization, C.M., F.G. and G.T.; formal Analysis, F.B., G.G. and R.M.; investigation, C.M. and F.B.; resources, R.M.; writing—original draft preparation, F.B. and G.T.; writing—review and editing, C.M., D.G. and F.G.; supervision, F.G.; funding acquisition, R.M. All authors have read and agreed to the published version of the manuscript.

**Funding:** This research received no external funding.

**Institutional Review Board Statement:** Not applicable.

**Informed Consent Statement:** Not applicable.

**Data Availability Statement:** Not applicable.

**Conflicts of Interest:** The authors declare no conflict of interest.

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
