# Peer review of "Effects of Starter Cultures and Type of Casings on the Microbial Features and Volatile Profile of Fermented Sausages"

_fermentation, doi:10.3390/fermentation8120683_

Round 1

Reviewer 1 Report

The paper entitled ‘’Effects of starter cultures and type of casings on the microbial features and volatile profile of fermented sausages’’ treats a very interesting and uninvestigated topic referring to the influence of three different commercial starter cultures and two types of sausage casing (natural and collagen) on the characteristics of Emiliano-type sausages (Italian fermented sausages).

Extensive explanations and correlation between the analyzed parameters are provided. The metabolic pathways of microbiota were taken into account in relationship with the chemicals identified at the end of sausages’ ripening. Further perspective of research could be underlined.

Reviewer 2 Report

Line 34. “fermented sausages are historically obtained stuffing minced lean and fat meat”. The sentence is quite confusing, please rewrite this sentence.

Line 172, please remove the word “mean” in the bracket.

Figure 1. please add information in x axis, about t0 (time, 0 days, etc..)

What is the role of internal and external pH in sausage making? Please discuss more briefly about this.

Table 1. please improve the quality of the table.

Replace t0 and t4 with day 0, day 4 respectively.

Do you have any reason, why you don’t determine the fatty acid profile and amino acid profile, as this will also contribute to the flavor formation.
